# Predictive value of physical and blood examination findings for short-term mortality in dogs with respiratory disorders

Muryo Miki[1,2,3], Keiichiro Mie[4], Hidetaka Nishida[5], Hideo Akiyoshi[6], Toshiyuki Tanaka[7]*

**1** Miki Animal Hospital, Kyoto, Kyoto, Japan, **2** Laboratory of Veterinary Surgery, Graduate School of Veterinary Science, Osaka Metropolitan University, Izumisano, Osaka, Japan, **3** Kyoto Animal Emergency Center, Kyoto, Kyoto, Japan, **4** Department of Comparative Animal Science, College of Life Science, Kurashiki University of Science and The Arts, Kurashiki, Okayama, Japan, **5** Laboratory of Small Animal Clinics, School of Veterinary Medicine, Azabu University, Sagamihara, Kanagawa, Japan, **6** Neovets VR center, Osaka, Osaka, Japan, **7** Laboratory of Veterinary Advanced Diagnosis and Treatment, Graduate School of Veterinary Science, Osaka Metropolitan University, Izumisano, Osaka, Japan

* f21724w@omu.ac.jp

## Abstract

### Background

Similar to human medicine, attempts have been made in veterinary medicine to assess the severity of respiratory disorders using methods other than respiratory function evaluation; however, such approaches remain insufficient.

### Measurements and main results

Medical records at a single small animal private referral center for emergency care were reviewed to identify dogs with respiratory disorders diagnosed by radiography during 2016–2019. The variables of screening test evaluated in this study included patient characteristics, physical examination, and blood test findings. The cases were also divided into Survivors, which were defined as dogs surviving over 7 days from the first consultation day, and Non-survivors, including the dogs euthanized and died naturally within 7 days from the consultation day. In univariate analysis, heart rates, body temperature, white blood count (WBC), glucose, blood urea nitrogen (BUN), phosphate and lactate were significantly different between Survivors and Non-survivors. Multiple logistic regression model with these significant variables revealed that only phosphate was associated with a poor prognosis.

### Conclusions

This study has demonstrated several parameters of physical examination and blood test, especially plasma phosphate concentration, could be related with mortality in canine respiratory disorders. Although further studies are needed, these parameters may enable more accurate assessment of the severity of respiratory disorders in

**Data availability statement:** All relevant data are within the manuscript and its Supporting Information files.

**Funding:** The author(s) received no specific funding for this work.

**Competing interests:** The authors have declared that no competing interests exist.

dogs by combining with the conventional assessments of respiratory functions including oxygenation and ventilation.

## Introduction

Respiratory disorders represent a frequent cause of emergency presentation in small animal veterinary practice [1]. These conditions often progress rapidly, leading to severe hypoxemia, and therefore require prompt and accurate initial assessment. Prolonged hypoxia may result in multi-organ dysfunction syndrome, highlighting the importance of systemic evaluation in addition to respiratory function assessment [2].

In human medicine, several scoring systems have been developed to assess the severity and prognosis of respiratory diseases using a combination of clinical parameters. For example, in community-acquired pneumonia, the Pneumonia Severity Index (PSI) has been established as a reliable scoring system with strong prognostic correlation [3]. PSI incorporates not only respiratory parameters but also readily available clinical data such as age, respiratory rate, blood pressure, and blood urea nitrogen (BUN). Multivariate analysis has shown that these systemic parameters are more strongly associated with outcomes than oxygenation and ventilation markers such as arterial blood gas values or $SpO_2$ [3]. Furthermore, it has been reported that patients with acute respiratory distress syndrome (ARDS) often succumb not to respiratory failure itself, but to ventilator-induced lung injury or multiple organ failure [4]. These findings emphasize the significance of systemic assessment in the prognostic evaluation of respiratory disorders.

Despite the widespread use of prognostic scoring systems in human respiratory medicine, veterinary medicine lacks analogous, disease-specific tools for assessing the prognosis of respiratory disorders. Prognostic evaluation is primarily based on respiratory function parameters such as $SpO_2$ and arterial blood gas analysis. General health assessment in dogs typically involves physical examination, blood tests, urinalysis, ultrasonography, and radiography; however, few studies have investigated the prognostic value of these data in canine respiratory disorders. For instance, a previous study reported that age, mental status, hypoalbuminemia, and elevated BUN were associated with outcomes in dogs with aspiration pneumonia [5]. Meanwhile, another study showed that total white blood cell count and neutrophil-to-lymphocyte ratio were not useful prognostic markers in canine pneumonia [6].

While efforts have been made to identify prognostic indicators beyond respiratory parameters, sufficient validation is lacking. Therefore, in this study, we hypothesized that abnormalities in commonly available physical and blood parameters would be associated with increased short-term mortality in dogs with radiographically confirmed respiratory disorders, and we aimed to investigate the feasibility of this approach.

## Materials and methods

### Criteria for case selection

Case records of a single small animal private referral center for emergency care were reviewed. Data from dogs presenting to the hospital between April 2016 and March

2019 were retrospectively surveyed for this study. Among dogs with clinical signs of respiratory abnormalities at presentation, the dogs confirmed abnormalities by craniocervical and thoracic radiography were defined as respiratory disorders for this study. The outcomes of the cases with respiratory diseases were surveyed by questionnaire to family veterinarians after discharge from our center. For the purposes of this study, the following cases were excluded: (a) cases without radiographic abnormalities in respiratory organs, (b) cases having no relationship between symptoms and radiographic findings, and (c) cases without follow-up data after 7 days from presentation. All radiographic findings were evaluated by the attending emergency veterinarian with at least 10 years of clinical experience, and the radiographic features were reviewed by first author.

## Study measures

According to previous study [7], these cases were classified as following disease locations identified by radiography; upper airways (including nose, larynx, pharynx, trachea, and main bronchi), lower airways (bronchi and bronchiole), lung parenchyma (vascular-interstitial-alveolar problems), and pleural space (including mediastinal masses and diaphragmatic problems). When the lesions were located at multiple parts, the cases were classified as multiple localizations. In this study, based on previous study, short-term prognosis was defined as 7 days, and dogs that survived more than 7 days from admission were classified as the Survivors, while those that died naturally or were euthanized within 7 days were classified as Non-survivors [8]. These groups were compared by patients characteristics (age, sex), physical examination (body weight, heart rate (HR), respiratory rate, body temperature (BT), Levine's grading of heart murmur) and blood test findings including WBC, packed cell volume (PCV), platelet (Plat), glucose (Glu), albumin, BUN, creatinine, calcium, phosphate, alanine aminotransferase, alkaline phosphatase, total bilirubin, total cholesterol, sodium, potassium, chloride, C-relative protein (CRP), lactate. Physical examination and collecting blood specimens were carried out on presentation. Blood test were performed using Procyte Dx Hematology Analyzer (IDEXX Laboratories Inc, Westbrook, ME), Catalyst One Chemistry Analyzer (IDEXX Laboratories Inc, Westbrook, ME) and Lactate Pro 2 LT-1730 (ARKRAY Marketing Inc, Tokyo).

## Statistical methods

Continuous variables were expressed as median (range) and compared with Mann Whitney's U-test. Categorical variables were analyzed using Fisher's exact test. For all statistical analyses, P value less than 0.05 was considered significant. A logistic regression model was constructed with outcome as the dependent variable. Independent variables were selected from the aforementioned univariate analyses with values of $P < 0.05$. To avoid model overfitting due to the limited sample size, the models were designed via backward elimination and optimized using the Akaike information criterion. Cases with missing values in any of the variables used for multivariate analysis were excluded using listwise deletion, in accordance with the default setting of the statistical software. Therefore, only complete cases were included in the logistic regression analysis. To investigate the utility of predictive ability of the independent variables that were statistically significant, cut-off value was calculated by Youden's Index analysis at receiver operating characteristic curves. Then, the sensitivity, specificity, positive and negative predictive value were calculated using that cut-off value. Data were recorded in an electronic spreadsheet and were analyzed using R, version 3.5.3. (veena, Austria)

## Ethics statement

Ethical approval was not required for this study as this is retrospective and utilized data from clinical patients. Furthermore, all data used in this study were obtained from the animal owners who agreed to the research application of clinical data in a pre-medical consent form, with due consideration given to personal information.

Euthanasia was carried out at the request and consent of the owner. Euthanasia was performed by intravenously administering propofol (10 mg/kg) followed by intravenously administering KCl (2 mEq K$^+$/kg).

## Result

Of 3010 dogs presented our hospital from April 2016 to March 2019, 225 dogs (7.5%) were diagnosed as respiratory disorders. The response rate to the follow-up questionnaire was 60.9% (137/225 cases). Four cases were excluded due to the lack of follow-up information from the referring veterinarians. Consequently, the remaining 133 dogs (59.1%) were included in the final analysis (Fig 1). The median age and body weight in dogs with respiratory disorders were 11.1 years (0.1–17 years) and 4.0 kg (0.4–29.2 kg), respectively. Chihuahua (40 cases; 35.4%), Miniature dachshund (16 cases; 12.0%), and Toy poodle (11 cases; 8.3%) were the most common breeds. There were 64 male dogs (48.1%), which included 44 intact cases (68.8% in male) and 20 castrated cases (31.3% in male), and 69 female dogs (51.9%), which included 28 intact cases (40.6% in female) and 41 spayed cases (59.4% in female). 105 dogs (78.9%) survived more than 7 days from presentation (Survivors) and 28 dogs (21.1%) died within 7 days (Non-survivors). Of Non-survivors, 27 dogs (96.4%) were died within 24 hours from presentation (including 1 euthanized dog), and 1 dog (3.6%) was died after 48–72 hours from presentation. The estimated causes of respiratory disorders are listed in Table 1. The localizations identified by X-ray test in Survivors were following: upper airways (27cases; 25.7%), lower airways (15 cases; 14.3%), lung parenchyma (47 cases; 44.8%), and pleural space (2 cases; 1.9%). The localizations in Non-survivors were following: upper airways (1 case; 3.6%), lung parenchyma (25 cases; 89.3%), and multiple parts (2 cases; 7.1%). Thus, lung parenchyma accounted for a large part of localizations both in Survivors and Non-survivors (Table 2). The patient characteristics, physical examination and laboratory findings of Survivors and Non-survivors are presented in Table 3. HR and BT in Non-survivors were significantly lower than Survivors, and WBC, Glu, BUN, phosphate, and lactate levels were significantly higher in Non-survivors than in Survivors. Multiple logistic regression model with these significant variables revealed that only phosphate was associated with a poor prognosis (Table 4). Other variables that were initially included in the model but excluded during the stepwise selection process are presented in S1 Table. Receiver operating characteristic curve analysis showed that phosphate concentration above 5.8 mg/dl was predictive for death with area under curve of 0.895 (Fig 2). Using this cut-off value, the sensitivity, specificity, positive and negative predictive value were 81%, 91%, 81%, and 91%, respectively (Table 5).

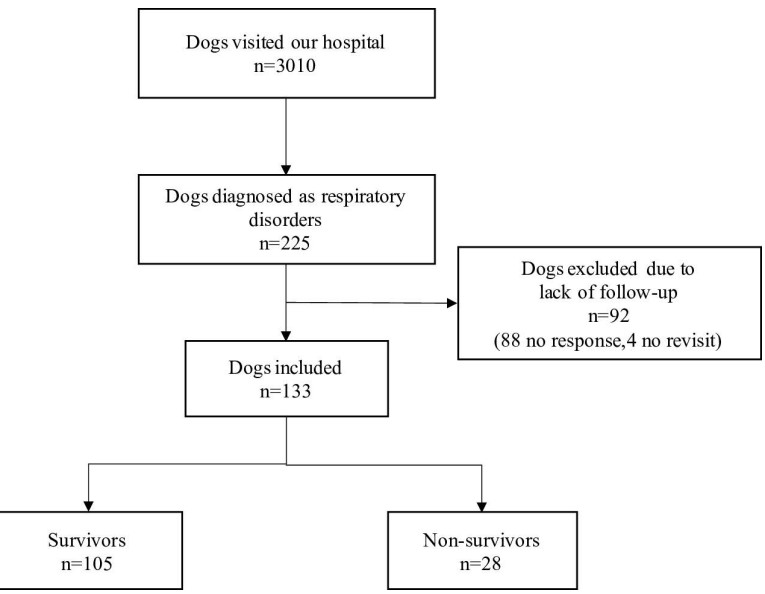

**Fig 1. Flow diagram of inclusion and exclusion of this study.** In addition to respiratory symptoms, dogs with radiographic abnormalities were diagnosed as respiratory disorders.

**Table 1. List of main estimated diagnoses for respiratory disorders.**

| Survivors (n = 105) | Non-survivors (n = 28) |
|---|---|
| Cardiogenic pulmonary edema (n = 37) | Cardiogenic pulmonary edema (n = 11) |
| Bronchitis (n = 23) | Pneumonia (n = 8) |
| Tracheal collapse (n = 16) | Acute respiratory distress syndrome (n = 4) |
| Pneumonia (n = 10) | Lung metastasis (n = 2) |
| Intranasal foreign body (n = 4) | Aspiratory pneumonia (n = 1) |
| Brachycephalic airway syndrome (n = 3) | Upper airway obstruction (n = 1) |
| Pulmonary hypertension (n = 2) | Lung contusion (n = 1) |
| Lung contusion (n = 2) | |
| Lung mass (n = 2) | |
| Pleural effusion (n = 2) | |
| Aspiration pneumonia (n = 1) | |
| Bronchiectasis (n = 1) | |
| Lung abscess (n = 1) | |
| Soft palate exaggeration (n = 1) | |

**Table 2. Localization of lesions in dogs with respiratory disorders.**

| Localization | Survivors | Non-survivors |
|---|---|---|
| Upper airways | 27 | 1 |
| Lower airways | 15 | 0 |
| Lung parenchyma | 42 | 25 |
| Pleural space | 2 | 0 |
| Multiple localizations | 19 | 2 |
| Total | 105 | 28 |

Similar statistical analysis was also performed only in the cases whose disorder was located in lung parenchyma because lung parenchyma was the majority of disorder sites both in Survivors and Non-survivors in this study. In univariate analysis, BT was significantly lower and WBC, Glu, phosphate were higher than Survivors (Table 6). In multivariate analysis, only phosphate was associated with a poor prognosis (Table 7). Although we didn't perform statistical analysis for other localizations due to insufficient number of cases, all cases of upper airways and multiple lesions in Non-survivors showed higher values than the cut-off value of phosphate in respiratory disorders. Furthermore, all cases of upper airways and multiple lesions in Survivors showed lower the phosphorus level than the cut-off value. This study included one euthanasia case, and even in this case the phosphorus level was higher than the cut-off value.

Since the results might have been influenced by the presence or absence of cardiac disease, a post hoc subgroup analysis was performed by dividing the cases into two groups: those with cardiogenic pulmonary edema (CPE), which is a common cause of respiratory distress in dogs, and those with other respiratory disorders (non-CPE). 48 cases were clinically suspected to have cardiogenic origin based on concurrent findings of heart murmur, radiographic features, echocardiographic parameters, or history of cardiac disease [9–11]. Univariate analysis showed that in dogs with CPE, BT was significantly lower, and Glu and phosphate concentration were significantly higher in Non-survivors than in Survivors (S2 Table). Multivariable logistic regression was attempted using these variables; however, due to a high rate of missing data in the Non-survivors, only 5 cases were available for analysis. In addition, strong collinearity was observed among the variables (VIF > 10). Although we applied stepwise selection using the Akaike information criterion to reduce the number of predictors, the model failed to converge, and multivariable analysis could not be performed. Therefore, only the results of

**Table 3. Comparisons of characteristics, physical examinations and laboratory values in survivor and non-survivor dogs with respiratory disorders.**

| Variable | n | Survivors | n | Non-survivors | P value |
|---|---|---|---|---|---|
| Age (years) | 105 | 11 (0.2 - 17) | 28 | 12.8 (0.1–15.5) | 0.481 |
| Sex (n) | 105 | male:54 female:51 | 28 | male:10 female:18 | 0.201 |
| BW (kg) | 105 | 3.94 (0.64 - 29.15) | 28 | 3.99 (0.433 - 16.6) | 0.830 |
| BCS (1–5) | 53 | 3 (1 - 5) | 16 | 3 (2 - 5) | 0.366 |
| Temperature (°C) | 98 | 38.5 (35.2 - 40.7) | 23 | 37.2 (34.1 - 40.3) | <0.001 |
| Heart rate (per min) | 69 | 142 (72 - 240) | 19 | 120 (60 - 192) | 0.031 |
| Respiratory rate (per min) | 55 | 78 (20 - 156) | 18 | 76 (30 - 156) | 0.728 |
| Cardiac murmur (1–6) | 94 | 0 (0 - 6) | 23 | 2 (0 - 5) | 0.370 |
| WBC (×10³/µL) | 74 | 13955 (3690 - 48390) | 26 | 20855 (6860 - 51740) | 0.002 |
| Platelets ((×10³/µL) | 74 | 368 (71 - 1173) | 26 | 408 (122 - 801) | 0.387 |
| PCV (%) | 73 | 43.4 (26.1 - 53.9) | 26 | 39.9 (23.7 - 70.5) | 0.867 |
| Glu (mg/dL) | 55 | 103 (75 - 243) | 26 | 151 (59 - 446) | <0.001 |
| Albumin (g/dL) | 58 | 3.1 (2.2 - 4.3) | 23 | 3 (2.1–4.0) | 0.247 |
| BUN (mg/dL) | 62 | 22 (7 - 67) | 25 | 28 (12 - 130) | 0.029 |
| Creatinine (mg/dL) | 62 | 0.9 (0.3 - 1.8) | 25 | 0.8 (0.3 - 5.5) | 0.534 |
| Calcium (mg/dL) | 43 | 9.6 (7.9 - 12.8) | 21 | 8.9 (7.3 - 12.0) | 0.081 |
| Phosphate (mg/dL) | 44 | 4.5 (2.8 - 8.5) | 21 | 6.7 (4.6 - 12.3) | <0.001 |
| ALT (U/L) | 56 | 71 (18 - 762) | 22 | 64 (26 - 182) | 1.000 |
| ALP (U/L) | 55 | 122 (10 - 981) | 23 | 114 (25 - 542) | 0.983 |
| Total bilirubin (mg/dL) | 39 | 0.2 (0.1 - 1.6) | 21 | 0.2 (0.1 - 3.4) | 0.623 |
| Total cholesterol (mg/dL) | 39 | 164 (79 - 247) | 22 | 184 (56 - 411) | 0.124 |
| Sodium(mEq/L) | 62 | 154 (147 - 163) | 25 | 154 (124 - 166) | 0.794 |
| Potassium(mEq/L) | 64 | 4.5 (3.1 - 5.5) | 25 | 4.5 (3.3 - 5.3) | 0.826 |
| Chloride(mEq/L) | 64 | 115 (105 - 125) | 25 | 114 (88 - 123) | 0.347 |
| CRP(mg/dL) | 75 | 2 (0-20) | 23 | 2.35 (0 - 20) | 0.609 |
| Lactate (mmol/L) | 14 | 4.1 (1.7–13.3) | 13 | 7.1 (1.9–19.8) | 0.013 |

**Table 4. Predictors for 7-day mortality in canine respiratory disorders by multivariate. logistic regression analysis.**

| Variable | Odds ratio | 95% CI | P value |
|---|---|---|---|
| Phosphate (mg/dL) | 2.38 | 1.02–5.59 | 0.046 |

Abbreviations: CI, confidence interval.

univariable analyses are reported for the CPE group. In contrast, among dogs in the non-CPE group, univariate analysis revealed that BT and Albumin were significantly lower and WBC, Glu, BUN and blood phosphate were significantly higher in Non-survivors compared to Survivors (S3 Table). Multivariable logistic regression showed only phosphate remained a statistically significant variable in the logistic regression analysis in non-CPE group (S4 Table). These findings were consistent with the results from the overall and lung parenchymal disorder groups.

## Discussion

This study demonstrated that several physical examination findings and blood test parameters were associated with prognosis in dogs with respiratory disorders. Multivariate analysis identified elevated plasma phosphate concentration as an

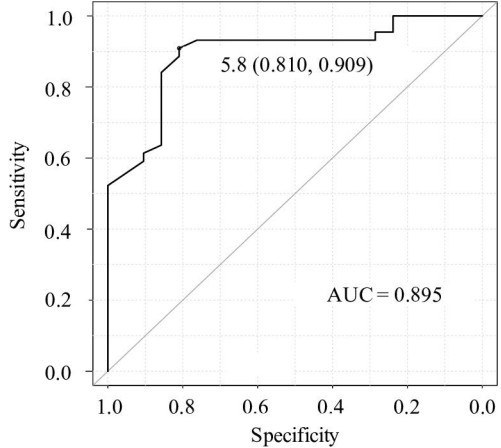

**Fig 2. ROC curve for plasma phosphate concentration as predictor for dogs with respiratory disorders.**

**Table 5. Performances of plasma phosphate concentration for predicting prognosis.**

| Variable | Cut-off value (mg/dL) | Sensitivity (%) | Specificity (%) | PPV (%) | NPV (%) |
|---|---|---|---|---|---|
| Phosphate | 5.8 | 81 | 91 | 81 | 91 |

Abbreviations: PPV, positive predictive value; NPV, negative predictive value.

**Table 6. Results of univariate analysis in dogs with lung parenchyma disorders.**

| Variable | n | Survivors | n | P value | Non-survivors |
|---|---|---|---|---|---|
| Temperature (°C) | 38 | 38.1 (35.2–40.7) | 20 | <0.001 | 37.3 (34.1–40.3) |
| WBC (×10³/μL) | 30 | 14060 (4370 - 48390) | 23 | 0.031 | 19400 (6860 - 51740) |
| Glu (mg/dL) | 25 | 103 (78 - 243) | 23 | 0.010 | 151 (59 - 446) |
| Phosphate (mg/dL) | 21 | 4.9 (3.0–8.5) | 18 | <0.001 | 6.6 (4.6–12.3) |

In this Table, variables associated with mortality (p < 0.05) are only shown.

**Table 7. Results of multivariate logistic regression analysis in dogs with lung parenchyma disorders.**

| Variable | Odds ratio | 95% CI | P value |
|---|---|---|---|
| Phosphate (mg/dL) | 3.45 | 1.57–7.59 | 0.002 |

independent prognostic indicator in canine respiratory disorders. This association remained significant even in subgroup analyses, suggesting phosphate may serve as a generalizable biomarker across different respiratory etiologies. While hyperphosphatemia is a known prognostic factor in chronic kidney disease and other critical conditions in both human and veterinary medicine [12–14], our findings highlight its potential role in acute respiratory disorder as well. Therefore, these parameters may enable more accurate assessment of the severity of respiratory disorders in dogs by combining with the conventional assessment of respiratory functions such as oxygenation and ventilation.

In human medicine, elevated phosphate levels have been linked to poor outcomes in diseases such as chronic obstructive pulmonary diseases and pneumonia, including those assessed using scoring systems like CURB-65 [15,16]. Similar

associations have been reported in critical illness and burn patients, where hyperphosphatemia often results from acute kidney injury and tissue breakdown [17,18]. Among various possible causes of elevated phosphate, circulatory insufficiency and transcellular shifts are likely the primary contributors in our cohort. Circulatory insufficiency is one of the main causes of prerenal AKI and metabolic acidosis, and sustained hypoxia induce death of cells. The observed decreases in body temperature and heart rate, along with elevated lactate levels in non-survivors, support the presence of circulatory insufficiency and severe hypoxia in these cases. Similarly, there are multiple reports that the non-survivors have developed into shock and exhibited significant decrease of urine output compared to the survivors in cases of human with pneumonia [19,20]; therefore, decrease of renal excretion and increase of transcellular shift of phosphate induced by circulatory insufficiency might cause increase of blood phosphate level in this study.

There are several causes of circulatory insufficiency in respiratory disorders, one of which is considered to be hypoxemia. It has been reported that severe drops of blood pressure and heart rates are developed after initial responses including hypertension and tachycardia under reduced oxygen condition in various animal species [21–27]. One of these studies has also demonstrated that the severity of the drop of body surface temperature, blood pressure and heart rates have been depending on concentration of oxygen [27]. The details of changes in hemodynamics induced by hypoxemia during the process of exacerbation of respiratory diseases are still unknown because these previous studies used healthy animals whose respiratory functions are normal. However, hypoxemia may also have a serious effect on hemodynamics in cases of respiratory disorders, and it is important to detect circulatory insufficiency when evaluating the prognosis of canine respiratory disorders.

There was the possibility that the change of phosphate metabolism has been related to hyperphosphatemia in this study, however, increases of blood parathyroid hormone and Fibroblast growth factor 23 level and decrease of blood 1,25(OH)$_2$ D level, which have reported to be the factors related with poor prognosis in several illness such as COPD and sepsis [28–31], have normally caused hypophosphatemia rather than hyperphosphatemia. Thus, although further study is needed, the elevation of plasma phosphate level in non-survivors in this study might not be caused by the change of phosphate metabolism. Unfortunately, while elevated phosphate levels were statistically associated with poor outcome, the underlying mechanisms remain speculative due to the lack of confirmatory data. This warrants further mechanistic studies.

While this study focused on plasma phosphate concentration as a prognostic indicator, elevated phosphate levels are also known to act as a pathophysiological factor that contributes directly to tissue injury. One well-established mechanism involves ectopic mineralization of soft tissues, including the cardiovascular system and kidneys [32]. Additionally, recent evidence suggests that hyperphosphatemia may directly damage pulmonary tissue by inducing oxidative DNA damage and promoting apoptosis in lung epithelial cells [33]. Although various therapeutic interventions for hyperphosphatemia, such as phosphate binders or dietary phosphorus restriction, are used in clinical practice, their efficacy in the context of acute respiratory disorders in dogs remains unclear. Therefore, further studies are needed to clarify not only the role of phosphate elevation in pulmonary injury, but also whether therapeutic management of hyperphosphatemia may improve outcomes in canine respiratory disorders.

There are a few reports about usefulness of physical examination and blood test data as a prognostic indicator for canine respiratory disorders as well as this study. For example, one study reported no significant difference in blood WBC counts between survivors and non-survivors in canine pneumonia, while another study identified ages, sex, BUN as prognostic indicators in canine aspiratory pneumonia, respectively [5,6]. In this study, WBC and BUN of Non-survivors were significantly higher than those of survivors by univariable analysis. In human medicine, it has been reported that Non-survivors with pneumonia have showed WBC counts increased significantly compared to the counts of Survivors during hospitalization [34,35]. There have also been some reports that elevation of BUN has been related with poor prognosis in human community-acquired pneumonia [36,37]. Considering the possibility of circulatory insufficiency and tissue injury related to hyperphosphatemia in Non-survivors in this study, elevation of BUN and WBC in Non-survivors might also

reflect severe condition including poor circulation and tissue damage at the time of measurement. Although a previous study reported that the variables including ages and sex were related with prognosis [5], these variables did not show significant different in this study. That prior research focused on three brachycephalic breeds (Pugs, French Bulldogs, and Bulldogs) with aspiration pneumonia and median age of dogs was 7 months (range: 2–163 months), which has been lower than the age of the cases in our study (132 months, range: 1–204 months). Although further studies are needed, variables such as ages and sex might only be a prognostic indicator in specific respiratory disease or specific breeds.

In this study, increased Glu was related with poor outcome by univariate analysis. There have been some reports that hyperglycemia has related with poor prognosis in human and canine critical illness including sepsis [38–40]. Insulin resistance induced by stress response including secretion of catecholamine, cortisol and inflammation cytokine has been considered as one of the causes of hyperglycemia in critical conditions [40]. Because Non-survivors in this study might have been in severe condition as described above, increased Glu of Non-survivors might also reflect the severe condition causing insulin resistance due to stress response.

To avoid model overfitting due to the limited sample size, we used a backward stepwise logistic regression model based on the Akaike Information Criterion to reduce the number of explanatory variables. Although variables such as BUN, WBC and Glu were significantly different in the univariate analysis, they were not retained in the multivariate model. This may reflect inter-variable correlations and the limited sample size, which reduced the statistical power for detecting independent associations. Therefore, in order to create a more accurate model, it is necessary to increase the number of samples and conduct further studies.

This study has several limitations. First, the study included only cases with radiographic evidence of respiratory disease, potentially excluding early-stage or functional disorders. However, this criterion was considered necessary measures to maintain diagnostic consistency and to enhance the reliability of comparative analysis. Second, the use of follow-up surveys to obtain clinical outcomes may introduce bias. The response rate was approximately 60%, which may have led to selection bias, as cases with incomplete follow-up data were excluded from the analysis. Since outcome data were not directly collected at the study institution, misclassification or underreporting of adverse outcomes cannot be completely ruled out. Future prospective studies with standardized follow-up protocols are warranted to minimize these biases. Third, although additional clinical variables such as respiratory pattern or effort of breathing were often assessed during examination, these parameters were inconsistently recorded and therefore excluded from statistical analysis. Prospective studies are warranted to evaluate their prognostic value. Fourth, the inclusion of euthanized dogs in Non-survivors may have introduced a confounding effect, as euthanasia decisions may have been influenced by factors other than clinical severity, such as owner preference or financial considerations. However, only one dog was euthanized in this study, and its plasma phosphate concentration exceeded the identified cut-off value, suggesting that the impact of euthanasia on the overall results was likely minimal. Finally, factors including sex hormones, which could affect to blood phosphate levels, could not evaluated in this study [15].

## Conclusions

To our knowledge, this is the first study to demonstrate that plasma phosphate concentration is associated with mortality in canine respiratory disorders. Although further studies are needed, parameters that showed significant differences in this study may enable more accurate assessment of the severity of respiratory disorders in dogs by combining with the conventional assessments of respiratory functions including oxygenation and ventilation.

## Supporting information

**S1 Table. Variables excluded during the stepwise logistic regression model selection process.** The table shows variables that were included at the beginning of the multivariate logistic regression model but were excluded during backward stepwise selection based on the Akaike information criterion. These values such as odds ratio and P value should be interpreted with caution, as they are model-dependent and do not reflect the final model output.
(DOCX)

**S2 Table. Univariate analysis between Non-survivors and Survivors in dogs with CPE.**
(DOCX)

**S3 Table. Univariate analysis between Non-survivors and Survivors in dogs with non-CPE group.**
(DOCX)

**S4 Table. Results of multivariate logistic regression analysis in dogs with non-CPE group.**
(DOCX)

## Author contributions

**Writing – original draft:** Muryo Miki.

**Writing – review & editing:** Keiichiro Mie, Hidetaka Nishida, Hideo Akiyoshi, Toshiyuki Tanaka.

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
