## [Decision Letter · Decision Letter 0]

Dear Dr. Tanaka,

Thank you for submitting your manuscript to PLOS ONE. After careful consideration, we feel that it has merit but does not fully meet PLOS ONE’s publication criteria as it currently stands. Therefore, we invite you to submit a revised version of the manuscript that addresses the points raised during the review process.

We look forward to receiving your revised manuscript.

Kind regards,

Tomasz W. Kaminski

Academic Editor

PLOS ONE

**Journal Requirements:**

Please ensure that your manuscript meets PLOS ONE's style requirements, including those for file naming. The PLOS ONE style templates can be found at https://journals.plos.org/plosone/s/file?id=wjVg/PLOSOne_formatting_sample_main_body.pdf and https://journals.plos.org/plosone/s/file?id=ba62/PLOSOne_formatting_sample_title_authors_affiliations.pdf 2. Thank you for including your ethics statement:  "N/A".    To comply with PLOS ONE submissions requirements, please provide the following information in the Methods section of the manuscript and in the “Ethics Statement” field of the submission form (via “Edit Submission”):  *  Please indicate whether an animal research ethics committee prospectively approved this research or granted a formal waiver of ethics approval.*  Please enter the name of your Institutional Animal Care and Use Committee (IACUC) or other relevant ethics board. Also include an approval number if one was obtained.*   If anesthesia, euthanasia, or any kind of animal sacrifice is part of the study, please include briefly in your statement which substances and/or methods were applied. For additional information about PLOS ONE submissions requirements for ethics oversight of animal work, please refer to http://journals.plos.org/plosone/s/submission-guidelines#loc-animal-research   Once you have amended this/these statement(s) in the Methods section of the manuscript, please add the same text to the “Ethics Statement” field of the submission form (via “Edit Submission”). 3. We note that your Data Availability Statement is currently as follows: All relevant data are within the manuscript and its Supporting Information files. Please confirm at this time whether or not your submission contains all raw data required to replicate the results of your study. Authors must share the “minimal data set” for their submission. PLOS defines the minimal data set to consist of the data required to replicate all study findings reported in the article, as well as related metadata and methods (https://journals.plos.org/plosone/s/data-availability#loc-minimal-data-set-definition). For example, authors should submit the following data: - The values behind the means, standard deviations and other measures reported;- The values used to build graphs;- The points extracted from images for analysis. Authors do not need to submit their entire data set if only a portion of the data was used in the reported study. If your submission does not contain these data, please either upload them as Supporting Information files or deposit them to a stable, public repository and provide us with the relevant URLs, DOIs, or accession numbers. For a list of recommended repositories, please see https://journals.plos.org/plosone/s/recommended-repositories. If there are ethical or legal restrictions on sharing a de-identified data set, please explain them in detail (e.g., data contain potentially sensitive information, data are owned by a third-party organization, etc.) and who has imposed them (e.g., an ethics committee). Please also provide contact information for a data access committee, ethics committee, or other institutional body to which data requests may be sent. If data are owned by a third party, please indicate how others may request data access. 4. When completing the data availability statement of the submission form, you indicated that you will make your data available on acceptance. We strongly recommend all authors decide on a data sharing plan before acceptance, as the process can be lengthy and hold up publication timelines. Please note that, though access restrictions are acceptable now, your entire data will need to be made freely accessible if your manuscript is accepted for publication. This policy applies to all data except where public deposition would breach compliance with the protocol approved by your research ethics board. If you are unable to adhere to our open data policy, please kindly revise your statement to explain your reasoning and we will seek the editor's input on an exemption. Please be assured that, once you have provided your new statement, the assessment of your exemption will not hold up the peer review process.

Reviewers' comments:

Reviewer's Responses to Questions

**Comments to the Author**

1. Is the manuscript technically sound, and do the data support the conclusions?

Reviewer #1: Yes

Reviewer #2: No

2. Has the statistical analysis been performed appropriately and rigorously?

Reviewer #1: Yes

Reviewer #2: No

3. Have the authors made all data underlying the findings in their manuscript fully available?

Reviewer #1: Yes

Reviewer #2: No

4. Is the manuscript presented in an intelligible fashion and written in standard English?

Reviewer #1: Yes

Reviewer #2: No

**Reviewer #1: ** In Findings of physical examination and blood test are related to poor prognosis in canine respiratory disorders: a retrospective evaluation of 113 dogs (2016-2019) by Miki et al the authors report on clinical factors of dogs with respiratory disorders. This study uses observational data collected from these dogs over a period of time and with matched outcomes.

I appreciate the straightforward and clear approach to this study. The authors used statistical methodology to identify phosphate, notable increased blood levels, as the predominant variable associated with negative outcomes within this clinical selected set of animals. The authors used appropriate methodology and analysis.

To help this study add additional context to these findings, I would like for the authors to expand on their discussion of the increased in phosphate; and more specifically, the therapeutic interventions that generally are used for hyperphosphatemia and how that may impact outcome. Or the authors could expand on the additional testing factors (i.e. hemoconcentrations) which could dive into a deeper understanding of the clinical mechanisms observed.

Overall, with this additional information providing some additional context to hyperphosphatemia I would like to suggest minor revisions before accepting this manuscript. I thank the authors for this clinical study and appreciate how it may benefit the prognostics of canine respiratory disease outcomes.

**Reviewer #2: ** This manuscript presents a retrospective study aimed at identifying physical and laboratory parameters associated with short-term mortality in dogs with radiographically diagnosed respiratory disorders. While the topic is clinically important and underexplored in veterinary medicine, the manuscript suffers from significant weaknesses in methodological design, statistical modeling, and the interpretation of results.

The manuscript requires extensive English editing.

Below is a detailed, line-by-line review of each section of the manuscript.

Title

The title is overly long and could be more concise

Abstract

Line 24 “Introduction” This line header is fine and serves its purpose, though unnecessary in an abstract

Lines 25–29 The objective is vague and lacks specificity. The phrase "utility of screening test as severity assessment" is unclear. Does the study aim to validate a prognostic score, or simply find statistically significant associations?

Introduction

Line 52 Opens with a generic statement lacking scientific focus. It would be stronger to immediately frame the importance of early prognostic indicators in acute respiratory presentations.

Lines 60-67 The reference to the PSI is a reasonable intuition, but the analogy from human medicine to canine patients is merely implied rather than substantiated. It is not clarified whether such a scoring tool exists in veterinary medicine, or whether this study aims to propose one.

Moreover, the statement that general clinical parameters are more predictive than measures of oxygenation is made without discussing the scope or validity of this claim (PSI is designed specifically for community-acquired pneumonia).

The authors should more clearly define the existing gap in veterinary medicine. Are similar prognostic tools already available? Have there been previous attempts to develop one? Is the purpose of this study to address a specific unmet need, or simply to explore associations between selected parameters?

Lines 79–81 The assertion that the usefulness of screening tests is unknown is overstated. There are multiple veterinary studies evaluating CBC/chemistry values as prognostic tools in critical care.

Lines 89-91 define a hypothesis: i.e., "We hypothesized that abnormalities in commonly available physical and blood parameters would be associated with increased short-term mortality in dogs with radiographically confirmed respiratory disease."

Materials and Methods

Lines 96–100 Selection criteria are vague. How was the diagnosis confirmed? Was there a standard diagnostic approach (e.g., radiographic criteria)? Was the diagnosis made by board-certified radiologists?

Lines 100–102 The use of a questionnaire sent to referring veterinarians post-discharge introduces recall and reporting bias. There’s no explanation of the response rate or how non-responders were handled.

Line 102-104 The exclusion criteria raise several concerns that should be addressed for clarity and methodological rigor. “Cases without radiographic abnormalities in respiratory organs” This suggests that only patients with visible radiographic changes were included. However, this may systematically exclude certain respiratory conditions (e.g., early-stage tracheal collapse, laryngeal paralysis, or functional respiratory disorders) where radiographs may appear normal. The authors should justify this decision and discuss the potential for selection bias.

“Cases having no relationship between symptoms and radiographic findings” This is vague and subjective. How was the “relationship” assessed? Who determined whether clinical signs matched radiographic findings? The criterion is unclear.

“Cases without follow-up data after 7 days” Excluding all cases without 7-day follow-up could introduce significant selection bias, especially if lost cases differ systematically from those retained (e.g., healthier animals discharged early or euthanized without veterinary follow-up). The authors should report how many cases were excluded for this reason and whether their baseline characteristics differed from included cases. Moreover, the grouping of dogs into “Survivors” and “Non-survivors” based on a 7-day window is arbitrary and not justified with clinical rationale or references. Please explain the rationale for excluding dogs without follow-up. This could introduce selection bias if dogs that recovered quickly or were lost to follow-up had different clinical characteristics.

Lines 118-119 The only physical examination findings considered were body weight, heart rate HR, respiratory rate, body temperature and heart murmur but detail on respiratory and cardiovascular clinical parameters at presentation are not provided (mucous membrane color, jugular vein distension or pulsation, pulse quality, respiratory pattern, effort of breathing, respiratory sounds, presence/absence of cough, orthopnea). In my opinion, for a study aimed at identifying prognostic indicators in canine respiratory disorders, the absence of these variables is a significant limitation.

Furthermore, the absence of any discussion regarding cardiac disease is a significant omission. Cardiogenic pulmonary edema is one of the most common causes of respiratory distress in dogs, and differentiating it from primary respiratory conditions is a critical aspect of case management. I am sure that cardiovascular disease was accurately ruled out. However, without this information, the reader cannot assess the case mix, nor understand the relevance of the prognostic indicators identified.

Line 129–139 The description of the statistical analysis is superficial. No mention of power calculation, model diagnostics, or handling of missing data.

Results

Line 163–165 96.4% of non-survivors died within 24 hours. This suggests the "Non-survivor" group is dominated by peracute presentations, making comparisons with survivors (who lived >7 days) potentially invalid.

Lines 174–176 Multivariate analysis yields phosphate as the only significant variable. In Table 4 please include OR and 95% CIs even if not retained in the final model.

Discussion

The discussion heavily references human studies without translating this effectively to veterinary contexts. There is an overreliance on speculative pathophysiology (e.g., transcellular phosphate shifts, AKI) without supportive data in this population.

Lines 240–261 The mechanism by which phosphate increases mortality is unclear. The authors speculate about AKI and tissue hypoxia, but provide no evidence of renal failure or acid-base disturbances in these dogs.

Lines 288–295 The discussion of BUN and WBC as correlates of poor outcome is consistent with critical care literature. However, the decision to exclude these from the final model should be discussed.

Lines 311–317 The potential confounding effect of euthanasia is acknowledged only minimally. There is no analysis separating dogs who died naturally from those euthanized.

Final recommendation

I appreciate the authors’ efforts to address a clinically relevant question. However, due to important methodological and clinical limitations, the manuscript is not suitable for publication in its current form.

**Do you want your identity to be public for this peer review?** For information about this choice, including consent withdrawal, please see our Privacy Policy

Reviewer #1: **Yes: ** Robert G. Schaut

Reviewer #2: No

---

## [Author Response · Author response to Decision Letter 1]

13 May 2025

Dr. Tomasz W. Kaminski

Academic Editor

PLOS ONE

May 14, 2025

Dear Dr. Kaminski,

Re: Resubmission of manuscript reference No. PONE-D-25-03766

Thank you for the opportunity to revise and resubmit our manuscript entitled

"Findings of physical examination and blood test are related to poor prognosis in canine respiratory disorders: a retrospective evaluation of 113 dogs (2016-2019)."

We appreciate the thoughtful and constructive feedback provided by the reviewers and editorial team. We have carefully addressed all comments and have substantially revised the manuscript accordingly.

Firstly, in accordance with the first comment of Reviewer 2, the title has been revised as "Predictive value of physical and blood examination findings for short-term mortality in dogs with respiratory disorders."

Secondly, we identified some inconsistencies in terminology and expressions within the main text and have made minor revisions to improve consistency, without altering the overall structure or conclusions of the study. In addition, typographical errors were found in Table 1,3,4, and 5. These errors have been corrected and are limited to the tables themselves, with no impact on the study results or subsequent statistical interpretations.

Below, we present each reviewer's comments and our response to it.:

Reviewer #1:

Comment: To help this study add additional context to these findings, I would like for the authors to expand on their discussion of the increased in phosphate; and more specifically, the therapeutic interventions that generally are used for hyperphosphatemia and how that may impact outcome. Or the authors could expand on the additional testing factors (i.e. hemoconcentrations) which could dive into a deeper understanding of the clinical mechanisms observed.

Response: Thank you for this insightful comment. We have expanded the Discussion section to include detailed descriptions of potential mechanisms by which elevated phosphate levels may contribute to tissue injury, including mineralization of soft tissues and oxidative DNA damage in lung epithelial cells. Furthermore, therapeutic interventions are also mentioned in the same paragraph.

Revised text (Lines 309–320): “While this study focused on plasma phosphate concentration as a prognostic indicator, elevated phosphate levels are also known to act as a pathophysiological factor that contributes directly to tissue injury. One well-established mechanism involves ectopic mineralization of soft tissues, including the cardiovascular system and kidneys [32]. Additionally, recent evidence suggests that hyperphosphatemia may directly damage pulmonary tissue by inducing oxidative DNA damage and promoting apoptosis in lung epithelial cells [33]. Although various therapeutic interventions for hyperphosphatemia, such as phosphate binders or dietary phosphorus restriction, are used in clinical practice, their efficacy in the context of acute respiratory disorders in dogs remains unclear. Therefore, further studies are needed to clarify not only the role of phosphate elevation in pulmonary injury, but also whether therapeutic management of hyperphosphatemia may improve outcomes in canine respiratory disorders.”

Reviewer #2:

Comment: The title is overly long and could be more concise.

Response: We appreciate the reviewer’s suggestions. As suggested, we have changed the title as " Predictive value of physical and blood examination findings for short-term mortality in dogs with respiratory disorders."

Comment: Line 24 “Introduction” This line header is fine and serves its purpose, though unnecessary in an abstract

Lines 25–29 The objective is vague and lacks specificity. The phrase "utility of screening test as severity assessment" is unclear. Does the study aim to validate a prognostic score, or simply find statistically significant associations?

Response: We appreciate these helpful suggestions. We have revised the abstract to better communicate our objectives to readers.

Revised text (Lines 25–28): “ Background. Similar to human medicine, attempts have been made in veterinary medicine to assess the severity of respiratory disorders using methods other than respiratory function evaluation; however, such approaches remain insufficient.”

Comment: Line 52 Opens with a generic statement lacking scientific focus. It would be stronger to immediately frame the importance of early prognostic indicators in acute respiratory presentations.

Response: We thank the reviewer for this constructive suggestion. In the revised Introduction, we have restructured the opening to more clearly highlight the clinical importance of early prognostic evaluation in dogs presenting with acute respiratory distress. We now begin by emphasizing the need for rapid and reliable risk stratification in veterinary emergency care.

Revised text (Lines 51–55): Respiratory disorders are among the most common causes of emergency presentations in small animal practice [1]. These conditions often progress rapidly, leading to severe hypoxemia, and therefore require prompt and accurate initial assessment. Prolonged hypoxia may result in multi-organ dysfunction syndrome, highlighting the importance of systemic evaluation in addition to respiratory function assessment [2].

Comment: Lines 60-67 The reference to the PSI is a reasonable intuition, but the analogy from human medicine to canine patients is merely implied rather than substantiated. It is not clarified whether such a scoring tool exists in veterinary medicine, or whether this study aims to propose one.

Moreover, the statement that general clinical parameters are more predictive than measures of oxygenation is made without discussing the scope or validity of this claim (PSI is designed specifically for community-acquired pneumonia).

The authors should more clearly define the existing gap in veterinary medicine. Are similar prognostic tools already available? Have there been previous attempts to develop one? Is the purpose of this study to address a specific unmet need, or simply to explore associations between selected parameters?

Lines 79–81 The assertion that the usefulness of screening tests is unknown is overstated. There are multiple veterinary studies evaluating CBC/chemistry values as prognostic tools in critical care.

Response: We appreciate this important point. We have revised the paragraph to explicitly state that while scoring tools like the Pneumonia Severity Index (PSI) are well-established in human medicine, there are currently no widely accepted equivalent scoring systems in veterinary respiratory medicine. We now clarify that our study does not aim to develop a scoring tool but rather to explore whether commonly obtained physical and blood examination findings are associated with short-term outcomes in dogs with radiographically confirmed respiratory disorders. We have also moderated the statement regarding general parameters versus oxygenation measures and clarified that this conclusion is based on prior PSI studies in humans, not intended to be directly generalized to veterinary patients.

Moreover, considering that previous studies have evaluated CBC and serum chemistry parameters in veterinary critical care settings, we have revised this sentence to reflect that while certain studies have investigated individual parameters, the prognostic value of a combination of physical and routine blood examination findings specifically in dogs with respiratory disorders remains unclear. This clarification has been added to better reflect the existing literature.

Revised text (Lines 68–77): “In contrast, veterinary medicine currently lacks disease-specific scoring systems for respiratory disorders. Prognostic evaluation is primarily based on respiratory function parameters such as SpO₂ and arterial blood gas analysis. General health assessment in dogs typically involves physical examination, blood tests, urinalysis, ultrasonography, and radiography; however, few studies have investigated the prognostic value of these data in canine respiratory disorders. For instance, a previous study reported that age, mental status, hypoalbuminemia, and elevated BUN were associated with outcomes in dogs with aspiration pneumonia [5]. Meanwhile, another study showed that total white blood cell count and neutrophil-to-lymphocyte ratio were not useful prognostic markers in canine pneumonia [6].”

Comment: Lines 89-91 define a hypothesis: i.e., "We hypothesized that abnormalities in commonly available physical and blood parameters would be associated with increased short-term mortality in dogs with radiographically confirmed respiratory disease."

Response: We appreciate this suggestion. As recommended, we have now explicitly stated our hypothesis at the end of the Introduction.

Revised text (Lines 68–77): “While efforts have been made to identify prognostic indicators beyond respiratory parameters, sufficient validation is lacking. Therefore, in this study, we hypothesized that abnormalities in commonly available physical and blood parameters would be associated with increased short-term mortality in dogs with radiographically confirmed respiratory disorders, and we aimed to investigate the feasibility of this approach.”

Comment: Lines 96–100 Selection criteria are vague. How was the diagnosis confirmed? Was there a standard diagnostic approach (e.g., radiographic criteria)? Was the diagnosis made by board-certified radiologists?

Lines 100–102 The use of a questionnaire sent to referring veterinarians post-discharge introduces recall and reporting bias. There’s no explanation of the response rate or how non-responders were handled.

Line 102-104 The exclusion criteria raise several concerns that should be addressed for clarity and methodological rigor. “Cases without radiographic abnormalities in respiratory organs” This suggests that only patients with visible radiographic changes were included. However, this may systematically exclude certain respiratory conditions (e.g., early-stage tracheal collapse, laryngeal paralysis, or functional respiratory disorders) where radiographs may appear normal. The authors should justify this decision and discuss the potential for selection bias.

“Cases having no relationship between symptoms and radiographic findings” This is vague and subjective. How was the “relationship” assessed? Who determined whether clinical signs matched radiographic findings? The criterion is unclear.

Response: In the revised Materials and Methods section, we have clarified that radiographic evaluations were performed by veterinarians with sufficient clinical experience. To ensure consistency in case selection, we included only cases with radiographic abnormalities and for which follow-up data were available. As the reviewer correctly pointed out, this criterion may have resulted in the exclusion of functional respiratory disorders or early-stage lesions with no visible radiographic changes. We have now addressed this potential for selection bias in the Limitations section. Additionally, we have now stated the number of cases excluded due to unavailable follow-up beyond 7 days in the Results section. Furthermore, we acknowledge that the use of questionnaires introduces potential sources of bias, such as recall and reporting bias, and we have now explicitly mentioned these in the Limitations section.

Revised text (Lines 96–98): “All radiographic findings were evaluated by the attending emergency veterinarian with at least 10 years of clinical experience, and the radiographic features were reviewed by first author.”

Revised text (Lines 358–362): “Firstly, only cases with radiographically confirmed respiratory abnormalities were included. As a result, certain conditions such as functional disorders or early-stage diseases without apparent radiographic changes may have been excluded. However, this criterion was considered necessary measures to maintain diagnostic consistency and to enhance the reliability of comparative analysis.”

Revised text (Lines 152–153): “The response rate to the follow-up questionnaire was 60.9 % (137/225 cases).”

Revised text (Lines 362–368): “Second, the use of follow-up surveys to obtain clinical outcomes may introduce bias. The response rate was approximately 60%, which may have led to selection bias, as cases with incomplete follow-up data were excluded from the analysis. Since outcome data were not directly collected at the study institution, misclassification or underreporting of adverse outcomes cannot be completely ruled out. Future prospective studies with standardized follow-up protocols are warranted to minimize these biases.”

“Cases without follow-up data after 7 days” Excluding all cases without 7-day follow-up could introduce significant selection bias, especially if lost cases differ systematically from those retained (e.g., healthier animals discharged early or euthanized without veterinary follow-up). The authors should report how many cases were excluded for this reason and whether their baseline characteristics differed from included cases. Moreover, the grouping of dogs into “Survivors” and “Non-survivors” based on a 7-day window is arbitrary and not justified with clinical rationale or references. Please explain the rationale for excluding dogs without follow-up. This could introduce selection bias if dogs that recovered quickly or were lost to follow-up had different clinical characteristics.

Line 163–165 96.4% of non-survivors died within 24 hours. This suggests the "Non-survivor" group is dominated by peracute presentations, making comparisons with survivors (who lived >7 days) potentially invalid.

Response: Regarding the 7-day follow-up period, we have clarified in the manuscript that our objective was to evaluate short-term prognosis in canine respiratory disorders. Following precedents from previous studies on acute and severe illnesses, we defined short-term outcome as survival within 7 days. Considering that the majority of non-survivors in our study died within 24 hours, we believe this classification is appropriate for capturing acute-phase outcomes.

Revised text (Lines 107–110):” In this study, based on previous study, short-term prognosis was defined as 7 days, and dogs that survived more than 7 days from admission were classified as the Survivors, while those that died naturally or were euthanized within 7 days were classified as Non-survivors [8].”

Revised text (Lines 153–155):” Four cases were excluded due to the lack of follow-up information from the referring veterinarians. Consequently, the remaining 133 dogs (59.1%) were included in the final analysis (Fig.1).”

Comment: Lines 118-119 The only physical examination findings considered were body weight, heart rate HR, respiratory rate, body temperature and heart murmur but detail on respiratory and cardiovascular clinical parameters at presentation are not provided (mucous membrane color, jugular vein distension or pulsation, pulse quality, respiratory pattern, effort of breathing, respiratory sounds, presence/absence of cough, orthopnea). In my opinion, for a study aimed at identifying prognostic indicators in canine respiratory disorders, the absence of these variables is a significant limitation.

Furthermore, the absence of any discussion regarding cardiac disease is a significant omission. Cardiogenic pulmonary edema is one of the most common causes of respiratory distress in dogs, and differentiating it from primary respiratory conditions is a critical aspect of case management. I am sure that cardiovascular disease was accurately ruled out. However, without this information, the reader cannot assess the case mix, nor understand the relevance of the prognostic indicators identified.

Response: We appreciate this important observation. As this was a retrospective study, not all physical examination variables (e.g., mucous membrane color, pulse quality, respiratory effort) were consistently recorded and thus were excluded from analysis. We have added this clarification to the revised manuscript. Regarding cardiac disease, we have now performed a subgroup analysis separating cases with cardiogenic pulmonary edema (CPE) from non-CPE, using clinical signs, radiographic features, echocardiographic findings (when availa

---

## [Decision Letter · Decision Letter 1]

Dear Dr. Tanaka,

Thank you for submitting your manuscript to PLOS ONE. After careful consideration, we feel that it has merit but does not fully meet PLOS ONE’s publication criteria as it currently stands. Therefore, we invite you to submit a revised version of the manuscript that addresses the points raised during the review process.

We look forward to receiving your revised manuscript.

Kind regards,

Tomasz W. Kaminski

Academic Editor

PLOS ONE

Journal Requirements:

Additional Editor Comments:

Dear Authors,

Thank you for your revised submission of the manuscript. As Reviewer 2 declined to assess the revision, I invited an additional expert (Reviewer 3) to provide an independent evaluation.

Reviewer 3 has recommended minor revisions, noting that the manuscript has improved meaningfully. I agree with this assessment - your revisions have addressed several important points raised previously, but a few aspects still require further clarification or adjustment before we can proceed toward a final decision.

Please carefully review the comments provided and revise your manuscript accordingly.

We appreciate your continued efforts and look forward to receiving your updated version.

Best regards,

Reviewers' comments:

Reviewer's Responses to Questions

**Comments to the Author**

Reviewer #1: All comments have been addressed

Reviewer #3: All comments have been addressed

2. Is the manuscript technically sound, and do the data support the conclusions?

Reviewer #1: Yes

Reviewer #3: Yes

3. Has the statistical analysis been performed appropriately and rigorously?

Reviewer #1: Yes

Reviewer #3: Yes

4. Have the authors made all data underlying the findings in their manuscript fully available?

Reviewer #1: Yes

Reviewer #3: Yes

5. Is the manuscript presented in an intelligible fashion and written in standard English?

Reviewer #1: Yes

Reviewer #3: Yes

Reviewer #1: (No Response)

Reviewer #3: Dear Authors,

It is my pleasure to review this interesting paper.

As I know, that this paper is after a few rounds of the revision I just wanna suggest few small changes in this paper.

In the introduction:

1. Suggestion 1: Clarify and tighten the opening sentence

Original: Respiratory disorders are among the most common causes of emergency presentations in small animal practice.

Suggested revision: Respiratory disorders represent a frequent cause of emergency presentation in small animal veterinary practice.

2. Suggestion 2: Strengthen the contrast between human and veterinary medicine

Original: In contrast, veterinary medicine currently lacks disease-specific scoring systems for respiratory disorders.

Suggested revision: Despite the widespread use of prognostic scoring systems in human respiratory medicine, veterinary medicine lacks analogous, disease-specific tools for assessing the prognosis of respiratory disorders.

In the discusion:

1. Clarify and streamline main findings (Opening paragraph)

Original: In this study, we demonstrated that several parameters of physical examination and blood test were related with prognosis.

Suggested revision: This study demonstrated that several physical examination findings and blood test parameters were associated with prognosis in dogs with respiratory disorders.

2. Eliminate redundancy and combine similar ideas (Phosphate findings)

You repeat the association between elevated phosphate and prognosis multiple times. Here’s a cleaner version of that section:

Revised summary paragraph (merging lines 253–259 and 263–265):

Multivariate analysis identified elevated plasma phosphate concentration as an independent prognostic indicator in canine respiratory disorders. This association remained significant even in subgroup analyses, suggesting phosphate may serve as a generalizable biomarker across different respiratory etiologies. While hyperphosphatemia is a known prognostic factor in chronic kidney disease and other critical conditions in both human and veterinary medicine, our findings highlight its potential role in acute respiratory disease as well.

3. Polish scientific phrasing and condense speculative language

Original: Although many causes of increasing blood phosphate level were considered in this study, circulatory insufficiency and transcellular shift might be the main causes of blood phosphate elevation.

Suggested revision: Among various possible causes of elevated phosphate, circulatory insufficiency and transcellular shifts are likely the primary contributors in our cohort.

4. Improve clarity in linking clinical signs to mechanisms

Original: Since tendency of decrease in body temperature, heart rates and increase in blood lactate concentration were observed in Non-Survivors, it was considered that circulatory insufficiency and severe hypoxia were occurred in Non-Survivors.

Suggested revision: The observed decreases in body temperature and heart rate, along with elevated lactate levels in non-survivors, support the presence of circulatory insufficiency and severe hypoxia in these cases.

5. Remove unnecessary repetition about human studies and phosphate

You repeat references to COPD, CURB-65, and phosphate in humans multiple times. Streamline with this:

Revised version:

In human medicine, elevated phosphate levels have been linked to poor outcomes in diseases such as COPD and pneumonia, including those assessed using scoring systems like CURB-65 [15,16]. Similar associations have been reported in critical illness and burn patients, where hyperphosphatemia often results from acute kidney injury and tissue breakdown [17,18].

6. Shorten and refine limitations section

Your limitations are well laid out but overly wordy. Here's a tighter version of the first point:

Original: Firstly, only cases with radiographically confirmed respiratory abnormalities were included. As a result, certain conditions such as functional disorders or early-stage diseases without apparent radiographic changes may have been excluded.

Suggested revision: First, the study included only cases with radiographic evidence of respiratory disease, potentially excluding early-stage or functional disorders.

All the best with your future research.

**Do you want your identity to be public for this peer review?** For information about this choice, including consent withdrawal, please see our Privacy Policy

Reviewer #1: **Yes: ** Robert G. Schaut

Reviewer #3: **Yes: ** Marta Wolosowicz

---

## [Author Response · Author response to Decision Letter 2]

4 Jul 2025

Dr. Tomasz W. Kaminski

Academic Editor

PLOS ONE

July 5, 2025

Dear Dr. Kaminski,

Re: Resubmission of manuscript reference No. PONE-D-25-03766

Thank you for the opportunity to revise and resubmit our manuscript entitled

" Predictive value of physical and blood examination findings for short-term mortality in dogs with respiratory disorders."

We thank you for your continued consideration of our manuscript and appreciate the constructive feedback from Reviewer 3. We are grateful for the reviewer’s thoughtful comments, which have helped us to improve the clarity and scientific rigor of our manuscript. Below we provide a point-by-point response to each of the suggestions.

Reviewer #3:

Comment: Clarify and tighten the opening sentence

Original: Respiratory disorders are among the most common causes of emergency presentations in small animal practice.

Suggested revision: Respiratory disorders represent a frequent cause of emergency presentation in small animal veterinary practice.

Response: We appreciate the reviewer’s suggestion to improve clarity in the opening sentence. As advised, we have revised the sentence to be more concise and direct.

Revised text (Lines 51–52): “Respiratory disorders represent a frequent cause of emergency presentation in small animal veterinary practice.”

Comment: Strengthen the contrast between human and veterinary medicine

Original: In contrast, veterinary medicine currently lacks disease-specific scoring systems for respiratory disorders.

Suggested revision: Despite the widespread use of prognostic scoring systems in human respiratory medicine, veterinary medicine lacks analogous, disease-specific tools for assessing the prognosis of respiratory disorders.

Response: Thank you for highlighting the need to better emphasize the contrast between human and veterinary medicine. We have incorporated the suggested revision to strengthen this comparison.

Revised text (Lines 68–70): “Despite the widespread use of prognostic scoring systems in human respiratory medicine, veterinary medicine lacks analogous, disease-specific tools for assessing the prognosis of respiratory disorders.”

Comment: Clarify and streamline main findings (Opening paragraph)

Original: In this study, we demonstrated that several parameters of physical examination and blood test were related with prognosis.

Suggested revision: This study demonstrated that several physical examination findings and blood test parameters were associated with prognosis in dogs with respiratory disorders.

Response: We thank the reviewer for pointing out the importance of clearly presenting the main findings. We have adopted the suggested wording to enhance clarity and readability.

Revised text (Lines 253–254): “This study demonstrated that several physical examination findings and blood test parameters were associated with prognosis in dogs with respiratory disorders.”

Comment: Eliminate redundancy and combine similar ideas (Phosphate findings)

You repeat the association between elevated phosphate and prognosis multiple times. Here’s a cleaner version of that section:

Revised summary paragraph (merging lines 253–259 and 263–265):

Multivariate analysis identified elevated plasma phosphate concentration as an independent prognostic indicator in canine respiratory disorders. This association remained significant even in subgroup analyses, suggesting phosphate may serve as a generalizable biomarker across different respiratory etiologies. While hyperphosphatemia is a known prognostic factor in chronic kidney disease and other critical conditions in both human and veterinary medicine, our findings highlight its potential role in acute respiratory disease as well.

Response: We agree with the reviewer that the section on phosphate findings was redundant. Accordingly, we have condensed and integrated the relevant content to avoid repetition and to improve logical flow.

Revised text (Lines 254–260): “Multivariate analysis identified elevated plasma phosphate concentration as an independent prognostic indicator in canine respiratory disorders. This association remained significant even in subgroup analyses, suggesting phosphate may serve as a generalizable biomarker across different respiratory etiologies. While hyperphosphatemia is a known prognostic factor in chronic kidney disease and other critical conditions in both human and veterinary medicine, our findings highlight its potential role in acute respiratory disorder as well.”

Comment: Polish scientific phrasing and condense speculative language

Original: Although many causes of increasing blood phosphate level were considered in this study, circulatory insufficiency and transcellular shift might be the main causes of blood phosphate elevation.

Suggested revision: Among various possible causes of elevated phosphate, circulatory insufficiency and transcellular shifts are likely the primary contributors in our cohort.

Response: Thank you for this insightful recommendation. We have modified the sentence to present a more scientifically appropriate and concise explanation, following your suggestion.

Revised text (Lines 268–270): “Among various possible causes of elevated phosphate, circulatory insufficiency and transcellular shifts are likely the primary contributors in our cohort.”

Comment: Improve clarity in linking clinical signs to mechanisms

Original: Since tendency of decrease in body temperature, heart rates and increase in blood lactate concentration were observed in Non-Survivors, it was considered that circulatory insufficiency and severe hypoxia were occurred in Non-Survivors.

Suggested revision: The observed decreases in body temperature and heart rate, along with elevated lactate levels in non-survivors, support the presence of circulatory insufficiency and severe hypoxia in these cases.

Response: We appreciate the reviewer’s effort to improve the clarity of our interpretation of the clinical signs. We have revised the sentence to better articulate the link between observed signs and the proposed pathophysiological mechanisms.

Revised text (Lines 271–274): The observed decreases in body temperature and heart rate, along with elevated lactate levels in non-survivors, support the presence of circulatory insufficiency and severe hypoxia in these cases.

Comment: Remove unnecessary repetition about human studies and phosphate

You repeat references to COPD, CURB-65, and phosphate in humans multiple times. Streamline with this:

Revised version:

In human medicine, elevated phosphate levels have been linked to poor outcomes in diseases such as COPD and pneumonia, including those assessed using scoring systems like CURB-65 [15,16]. Similar associations have been reported in critical illness and burn patients, where hyperphosphatemia often results from acute kidney injury and tissue breakdown [17,18].

Response: We acknowledge the repetition in referencing human studies related to phosphate. We have revised the section to present these findings more concisely, as suggested.

Revised text (Lines 264–268): “In human medicine, elevated phosphate levels have been linked to poor outcomes in diseases such as chronic obstructive pulmonary diseases and pneumonia, including those assessed using scoring systems like CURB-65 [15,16]. Similar associations have been reported in critical illness and burn patients, where hyperphosphatemia often results from acute kidney injury and tissue breakdown [17,18].”

Comment: Shorten and refine limitations section

Your limitations are well laid out but overly wordy. Here's a tighter version of the first point:

Original: Firstly, only cases with radiographically confirmed respiratory abnormalities were included. As a result, certain conditions such as functional disorders or early-stage diseases without apparent radiographic changes may have been excluded.

Suggested revision: First, the study included only cases with radiographic evidence of respiratory disease, potentially excluding early-stage or functional disorders.

Response: We appreciate the reviewer’s feedback on improving the clarity and conciseness of the limitations section. We have rephrased the initial point in line with your recommendation.

Revised text (Lines 350–352): “First, the study included only cases with radiographic evidence of respiratory disease, potentially excluding early-stage or functional disorders.”

We have carefully addressed all comments and incorporated the suggested revisions into the manuscript. All changes have been clearly highlighted in the revised version. We hope that these revisions meet your expectations and that our manuscript is now suitable for publication in PLOS ONE. Thank you once again for your time and the constructive feedback.

Sincerely,

Toshiyuki Tanaka, DVM, PhD

Osaka Metropolitan University

Laboratory of Veterinary Advanced Diagnosis and Treatment

1-58, Rinku-oraikita, Osaka city, Osaka, 598-8531, Japan

email: f21724w@omu.ac.jp

---

## [Editor Report · Decision Letter 2]

Predictive value of physical and blood examination findings for short-term mortality in dogs with respiratory disorders

PONE-D-25-03766R2

Dear Dr. Tanaka,

We’re pleased to inform you that your manuscript has been judged scientifically suitable for publication and will be formally accepted for publication once it meets all outstanding technical requirements.

Kind regards,

Tomasz W. Kaminski

Academic Editor

PLOS ONE

---

## [Editor Report · Acceptance letter]

PONE-D-25-03766R2

PLOS ONE

Dear Dr. Tanaka,

I'm pleased to inform you that your manuscript has been deemed suitable for publication in PLOS ONE. Congratulations! Your manuscript is now being handed over to our production team.

Kind regards,

on behalf of

Dr. Tomasz W. Kaminski

Academic Editor

PLOS ONE